# Calorie and nutrient trends in large U.S. chain restaurants, 2012-2018

**Sara N. Bleich**[1]*, **Mark J. Soto**[1], **Caroline Glagola Dunn**[1], **Alyssa J. Moran**[2], **Jason P. Block**[3]

**1** Department of Health Policy and Management, Harvard T.H. Chan School of Public Health, Boston, MA, United States of America, **2** Department of Health Policy and Management, Johns Hopkins Bloomberg School of Public Health, Baltimore, MD, United States of America, **3** Department of Population Medicine, Harvard Medical School, Boston, MA, United States of America

* sbleich@hsph.harvard.edu

**Data Availability Statement:** The data used for this analysis are third party data. Others can access the data from the MenuStat Project (www.menustat.org). The authors did not have any special privileges that others would not have.

## Abstract

### Introduction

Large chain restaurants reduced calories in their newly-introduced menu items from 2012 to 2015. The objective of this study was to provide updated calorie trends through 2018 and examine trends in the macronutrient composition of menu items across this time period.

### Methods and findings

Data were obtained from the MenuStat project and include 66 of the 100 largest revenue generating U.S. chain restaurants (N = 28,238 items) that had data available in all years from 2012 to 2018. Generalized linear models were used to examine per-item calorie and nutrient changes (saturated fat, trans fat, unsaturated fat, sugar, non-sugar carbohydrates, protein, sodium) among (1) items on the menu in all years (common items) and (2) newly introduced items (2013–2018). Overall, there were no significant changes in calories or nutrients among common items from 2012 to 2018. Among all newly introduced items, calories (-120 kcals, -25%, p = 0.01; p-for-trend = 0.02), saturated fat (-3.4g, -41%, p<0.01, p-for-trend = 0.06), unsaturated fat (-4.5g, -37%, p = 0.02; p-for-trend = 0.04), non-sugar carbohydrates (-10.3g, -40%, p = 0.02, p-for-trend = 0.69), and protein (-4.3g, -25%, p = 0.04, p-for-trend = 0.02) declined.

### Conclusion

Newly introduced menu items in large chain restaurants have continued to decline in calories through 2018, which may help to reduce calorie intake. Other changes in macronutrient content were sporadic and not clearly toward improved dietary quality.

**Funding:** The authors received no specific funding for this work.

**Competing interests:** Caroline Dunn owns stock over a value of $5,000 in Beyond Meat, Inc. The other authors have declared that no competing interests exist. This does not alter our adherence to PLOS ONE policies on sharing data and materials.

## Introduction

Over half of food expenditures in the United States are spent on food away from home, a majority (72%) of which is at restaurants.[1] In 2019, Americans are expected to have spent over $863 billion at over 1 million restaurants across the U.S.[2, 3] Compared to eating at home, eating in restaurants is associated with consuming more calories, fat, sodium, and cholesterol[4] as well as increased risk for obesity.[5–8]

In recent years, large chain restaurants have made calorie and nutrient changes to menu items, which may benefit population health. From 2012 to 2014, the calorie content of newly introduced menu items decreased by about 60 calories (or 12 percent) [9]. From 2012 to 2015, mean calories for items that remained on the menu were significantly lower than mean calories for items that were dropped from the menu.[9] Among fast food restaurants, this decline was driven primarily by a decrease in calories from unsaturated fat.[10] From 2012 to 2016, calorie-adjusted sodium content in newly introduced menu items declined by 104mg.[11]

Given the size of the restaurant industry, the frequency with which Americans eat out, and the poor nutritional quality of restaurant food, and the resulting adverse health consequences of overconsumption, continued surveillance of calorie and nutrient changes to restaurant menu offerings can provide important information about ongoing trends. MenuStat is a database that tracks nutritional content of menu items, currently through 2018, in nearly all of the 100 top-selling U.S restaurant chains, representing about half of all US restaurant revenue [12]. This updated database provides an opportunity to expand prior analyses with data through 2018 and to compare across a 7-year time period.[13] [12] We used the MenuStat data from 2012 to 2018 to examine trends in calories and nutrients on the menu of large U.S. chain restaurants. Building on prior research, we hypothesized that calories and nutrients among items commonly on the menu each year remained the same and that calories and nutrients most associated with chronic disease risk (sugar, saturated fat, sodium) decreased among newly introduced items, with some variability across menu categories.

## Methods

### Data

We obtained data from the MenuStat Project (menustat.org), provided by the New York City Department of Health and Mental Hygiene. The Menustat team annually collects calorie and nutrient information about menu items made public by restaurants on their websites. Menu items are assigned to one of 12 mutually exclusive menu categories, and a unique, numeric identifier is assigned to each item (see **S1 File** for details). For items that appear on menus in multiple years, the menu ID remains the same. We restricted our analysis to 66 fast food, fast casual, and full service restaurant chains that appear in all seven years of data (2012–2018), which includes N = 28,238 unique menu items (see **S1 Table** for details). Data were collected for 2012–2018 and analyzed in 2019.

### Measures

We examined two sets of continuous outcomes: 1) mean, within-item changes in calories and nutrients (e.g., sodium, sugar and saturated fat) from 2012 to 2018 among items on the menu in all years (hereafter referred to as common items) and 2) mean, between-item calories and nutrients among newly introduced items from 2013 to 2018. We were particularly interested in changes in sodium, sugar, and saturated fat, and display those results graphically, as they are nutrients of high public health importance. Consistent with prior studies using these data,[9–11, 14–16] menu items were identified as common if they had the same name, item

description, and item ID in all years; items were defined as newly introduced in 2013–2018 if they did not have a matching name, item description, or item ID in any preceding year.

The primary outcomes were calories and nutrients in common items and newly introduced items, measured annually. We included restaurant-level covariates: restaurant saturation across the U.S. (whether the restaurant chain was located in all 9 Census divisions vs. not) and restaurant type (fast food, full service, fast casual). Given that many large chain restaurants voluntarily implemented menu labeling in advance of the federal rule and during the study period,[17] we additionally included an indicator for the year the restaurant began labeling their menus with calories. More information about these covariates can be found in **S1 File**, **S1 Table**, and **S2 Table**.

Information about calories and nutrients (saturated fat, trans fat, unsaturated fat, sugar, non-sugar carbohydrates, protein, and sodium) were available for most items from 2012–2018 (see **S3 Table** for missingness by nutrient type). Items were excluded from the analysis if they: 1) were common items that lacked a measure for calories in any year of data (N = 389; 1%), 2) were newly introduced items that lacked calorie information in the year they were introduced (N = 7,636; 15%), or 3) were on a menu in 2012 and removed in any year prior to 2018 (N = 10,805; 21%). We removed these latter items because we could not determine if they were newly introduced; we did not have data before 2012 to know if they appeared previously on the menu or were new in 2012. For an item missing data for any additional nutrient of interest, the item was excluded only from the analysis for that nutrient (e.g., saturated fat) in the year the datum was missing but included in all other analyses. We also excluded the toppings and ingredients category because restaurant reporting of these items was inconsistent over the study period. In some cases, toppings and ingredients were "double reported" in the MenuStat database, in that they were accounted for within certain types of items and also reported separately as stand-alone ingredients (e.g., pepperoni was added as a new topping in 2017 although pepperoni pizza had been included on the menu in all prior years of the data).

## Statistical analysis

We conducted all analyses in 2019 using Stata Version 15 (StatCorp LLC, College Station, TX), using generalized linear models to examine per-item trends across all years for common and newly introduced items as well as comparing 2012 vs. 2018 for common items and 2013 vs. 2018 for newly introduced items. We accounted for clustered observations at the restaurant chain level because items within chains may have been correlated, meaning that menu items within a restaurant chain may be more similar than menu items between restaurant chains. The unit of observation was the item and the primary outcome was calories or nutrient value of interest. We controlled for the covariates described above, and the margins command was used to estimate the yearly predicted mean per-item nutrient values for each menu category and sub-category (e.g., all items; items categorized as food or beverage; food subcategorized as appetizer or side, main dish, etc.) post-regression.

For each set of outcomes (calories and nutrients in common items and newly introduced items), we examined two models. First, to test for a linear trend, the main independent variable was a continuous variable for the year the item was offered (common items) or introduced (newly introduced items). Second, to compare differences between 2012 vs. 2018 (common items) or 2013 vs. 2018 (newly introduced items), year was specified as a categorical variable. For both models, we included item-level covariates (children's menu item status, regional, limited-time offers) and whether the item is categorized as shareable (also a MenuStat designation) which is similar to prior publications using the MenuStat data,[9–11, 14, 15, 18]. To account for correlated observations over time for common items only, we also included item

fixed effects and removed all time-invariant restaurant-level covariates. For relevant models, we report either p-values for trend to indicate linear changes over time or p-values to indicate changes between the first and last year of the data.

We conducted several sensitivity analyses to test the robustness of our main findings. First, we adjusted the nutrient analyses for the calorie content of menu items to determine whether observed changes in nutrients were independent of changes in calories. Second, for the nutrient analyses, we excluded items when calories were ±50 calories different than the sum of their nutrient calories (N = 1149, 5%) or when calories could not be calculated from nutrients due to missing information about those nutrients (N = 559, 2%). Third, we excluded items from restaurants that did not label their menus with calories by 2018 as required by federal law (N = 338, 1%). Fourth, we classified items as shareable that were not identified as such by the MenuStat team but had >2000 calories (N = 24, 0.1%) or >1000 calories for appetizers & sides (N = 52, 0.2%).

## Results

### Characteristics of menu items

Table 1 shows the characteristics of menu items in the 66 chain restaurants from 2012 to 2018, overall and by category. Of the 28,238 items, 48% were in fast food restaurants, 54% were food items (most in the main course category), 46% were beverages, and 6% were children's items. Overall, 20% of food items and 10% of beverage items were common and appeared on the menu in all years. In each year 2013 to 2018, newly introduced items represented 12% to 20% of all food items and 10% to 25% of all beverage items. Additional characteristics of menu items in the 66 chain restaurants can be found in S2 Table.

Table 1. Characteristics of items on menus in 66 U.S. restaurants from 2012–2018, overall and by category.

| Menu Category | | On menus in all years | Items new in 2013 | Items new in 2014 | Items new in 2015 | Items new in 2016 | Items new in 2017 | Items new in 2018 |
|---|---|---|---|---|---|---|---|---|
| **Overall (n)** | 28,238 | 4,284 | 4,146 | 3,328 | 3,551 | 3,747 | 5,203 | 3,979 |
| Food[a] | 54.4% | 71.2% | 48.4% | 53.2% | 55.7% | 65.9% | 37.8% | 53.3% |
| Beverages | 45.6% | 28.8% | 51.6% | 46.8% | 44.3% | 34.1% | 62.2% | 46.7% |
| **Food Category (n)** | 15,357 | 3,049 | 2,006 | 1,769 | 1,977 | 2,469 | 1,965 | 2,122 |
| Appetizers & sides | 11.2% | 11.8% | 6.0% | 9.8% | 10.2% | 13.5% | 12.5% | 13.6% |
| Main courses | 70.5% | 61.3% | 75.6% | 72.2% | 72.0% | 70.9% | 72.9% | 73.5% |
| Fried Potatoes | 2.4% | 5.3% | 1.9% | 1.5% | 2.6% | 0.6% | 2.0% | 1.3% |
| Desserts & baked goods | 15.9% | 21.6% | 16.5% | 16.5% | 15.2% | 15.0% | 12.5% | 11.7% |
| **Main course category (n)** | 10,829 | 1,869 | 1,516 | 1,278 | 1,424 | 1,750 | 1,433 | 1,559 |
| Burgers | 8.4% | 11.8% | 7.2% | 7.2% | 7.9% | 7.2% | 8.0% | 8.8% |
| Entrees | 37.5% | 33.9% | 29.9% | 39.5% | 45.4% | 43.5% | 33.6% | 37.3% |
| Pizza | 14.7% | 11.3% | 7.7% | 8.2% | 13.3% | 12.2% | 26.9% | 23.7% |
| Salads | 8.4% | 8.1% | 9.1% | 10.4% | 7.7% | 8.9% | 8.4% | 6.5% |
| Sandwiches | 26.8% | 26.9% | 38.3% | 32.0% | 23.7% | 26.1% | 19.7% | 21.6% |
| Soups | 4.2% | 8.1% | 7.9% | 2.7% | 2.0% | 2.2% | 3.3% | 2.1% |

Major row values are *n* of all menu items. Minor row values are proportions (%) of the major row column totals, unless otherwise indicated.

[a] Included all menu categories except beverages and toppings & ingredients

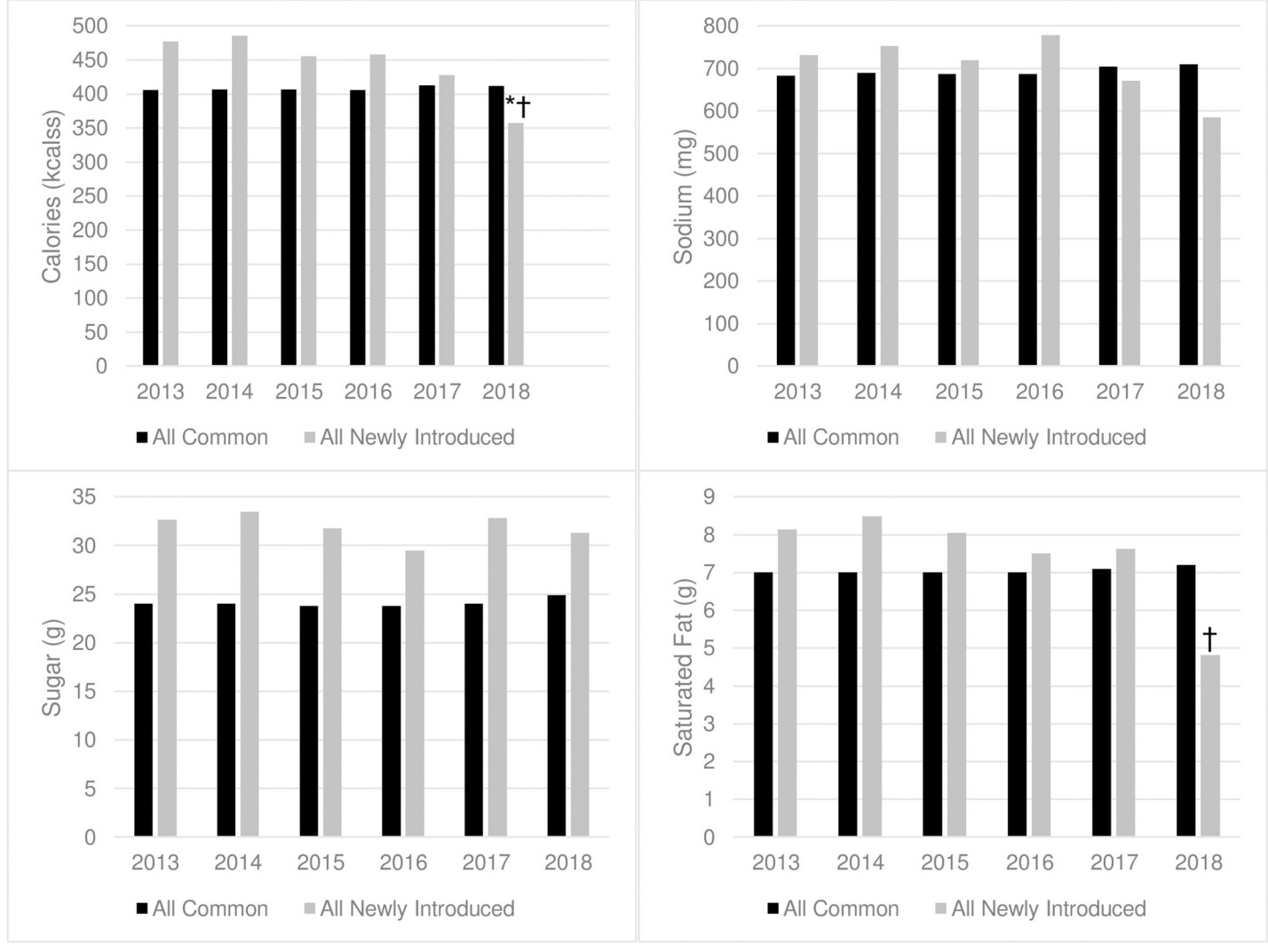

**Fig 1. Predicted per-item means for common and newly introduced items, 2012–2018.** All estimates are adjusted for restaurant type, whether the restaurant is a national chain, the year the restaurant began labeling their menus with calories, and whether the item is categorized as a kid's item, shareable, regional or offered for a limited time. Estimates for common items included item fixed effects. *The estimate for the trend from 2013 to 2018 is significant at p<0.05. †The estimate for the change from 2013 to 2018 is significant at p<0.05.

## Predicted means among common and newly introduced items

Fig 1 shows predicted means among common and newly introduced items (combining food and beverages together) for calories and several nutrients of particular public health importance (sodium, sugar, and saturated fat). There was no significant change in calories or nutrients among common items. For newly introduced items, calories (-120 kcals, -25%, p = 0.01; p-for-trend = 0.02) and saturated fat (-3.4g, -41%, p<0.01, p-for-trend = 0.06) declined (S4 Table). Also among newly introduced items, unsaturated fat (-4.5g, -37%, p = 0.02; p-for-trend = 0.04), non-sugar carbohydrates (-10.3g, -40%, p = 0.02, p-for-trend = 0.69), and protein (-4.3g, -25%, p = 0.04, p-for-trend = 0.02) declined (S4 Table). For each year prior to 2018, the mean predicted calories among newly introduced items was higher than mean calories among common items. However, in 2018, the mean calories and saturated fat in newly introduced items was lower than common items. Sodium in newly introduced items has

**Table 2. Predicted mean per-item calories, saturated fat, trans fat, unsaturated fat, sugar, non-sugar carbohydrates, protein and sodium for common food and beverage items on menus in 2012–2018.**

| Menu Category | n | Means | | | | | | | p-value for trend | 2012–2018 | |
|---|---|---|---|---|---|---|---|---|---|---|---|
| | | 2012 | 2013 | 2014 | 2015 | 2016 | 2017 | 2018 | | Change | p-value |
| **Food**[a] | | | | | | | | | | | |
| Calories (kcals) | 3049 | 452 | 460 | 463 | 463 | 461 | 472 | 467 | 0.14 | 15 kcals | 0.22 |
| Saturated fat (g) | 2963 | 8.3 | 8.4 | 8.3 | 8.4 | 8.4 | 8.6 | 8.6 | 0.17 | 0.3 g | 0.20 |
| Trans fat (g) | 2425 | 0.34 | 0.36 | 0.36 | 0.33 | 0.33 | 0.33 | 0.29 | 0.12 | -0.06 g | 0.11 |
| Unsaturated fat (g) | 2425 | 13.6 | 13.5 | 13.5 | 13.5 | 13.5 | 13.6 | 13.5 | 0.99 | -0.1 g | 0.72 |
| Sugar (g) | 2433 | 12.4 | 12.3 | 12.4 | 12.2 | 12.0 | 12.4 | 12.9 | 0.72 | 0.5 g | 0.58 |
| Non-sugar carbohydrates (g) | 2428 | 29.3 | 29.3 | 29.7 | 29.7 | 30.5 | 30.5 | 29.5 | 0.24 | 0.2 g | 0.68 |
| Protein (g) | 2992 | 18.5 | 19.0 | 18.9 | 18.8 | 18.6 | 19.0 | 18.8 | 0.87 | 0.3 g | 0.65 |
| Sodium (mg) | 2974 | 888 | 916 | 926 | 922 | 921 | 948 | 953 | 0.14 | 65 mg | 0.16 |
| **Beverages** | | | | | | | | | | | |
| Calories (kcals) | 1235 | 273 | 272 | 270 | 270 | 269 | 268 | 275 | 0.83 | 2 kcals | 0.67 |
| **Saturated fat (g)** | **1221** | **3.8** | **3.7** | **3.7** | **3.7** | **3.6** | **3.6** | **3.6** | **0.02** | -0.2 g | 0.10 |
| Trans fat (g) | 1098 | 0.09 | 0.10 | 0.11 | 0.12 | 0.12 | 0.11 | 0.11 | 0.30 | 0.02 g | 0.22 |
| Unsaturated fat (g) | 1098 | 2.2 | 2.2 | 2.1 | 2.1 | 2.1 | 2.1 | 2.1 | 0.13 | -0.1 g | 0.22 |
| Sugar (g) | 1096 | 50.1 | 50.1 | 49.7 | 49.5 | 49.9 | 49.7 | 51.5 | 0.46 | 1.4 g | 0.26 |
| Non-sugar carbohydrates (g) | 1081 | 4.4 | 4.4 | 4.3 | 4.4 | 4.6 | 4.6 | 4.8 | 0.33 | 0.4 g | 0.26 |
| Protein (g) | 1172 | 4.1 | 4.1 | 4.1 | 4.1 | 4.1 | 4.0 | 4.0 | 0.25 | -0.1 g | 0.13 |
| Sodium (mg) | 1232 | 123 | 119 | 119 | 119 | 120 | 120 | 122 | 0.98 | -1 mg | 0.88 |

Boldface indicates statistical significance at $p<0.05$. The n indicates total number of items available in all years for that category. All estimates included item fixed effects and are adjusted for whether the item is shareable, regional or offered for a limited time.

[a] Included all menu categories except beverages and toppings & ingredients.

remained below common items since 2017. The mean sugar in newly introduced items remained consistently higher than common items in all years of data.

## Predicted mean per-item calories and nutrients for common food and beverage items

Table 2 shows the annual predicted mean per-item calories and nutrients for common food and beverage items. The only significant change was a declining trend for saturated fat among beverages (-0.2g, -5%, 2018 vs. 2013 p = 0.10, p-for-trend = 0.02). Within sub-categories of common items, significant changes were observed for appetizers (sodium increased, +112mg, 17%, p = 0.01, p-for-trend = 0.01; sugar decreased -0.7g, -12%, p = 0.04, p-for-trend = 0.03), and fried potatoes (sodium increased +110mg, 17%, p = 0.04, p-for-trend = 0.04) (S5 Table).

## Predicted mean per-item calories and nutrients among newly introduced food and beverage items

Table 3 shows predicted mean per-item calories and nutrients among food and beverage items newly introduced to the menu from 2013 to 2018, the change in the nutrient of interest from 2013 to 2018, the p-value for the 2013 to 2018 change, and the p-value for the trend from 2013 to 2018. Among newly introduced food items, sugar (-6.7g, -35%, p = 0.02, p-for-trend = 0.01) declined. Among newly introduced beverages, protein (-2.7g, -44%, p<0.01, p-for-trend< 0.01) declined across the study period and when comparing values from the first and last years of data. Unsaturated fat (-1.8g, -58%, p = 0.01, p-for-trend = 0.23) and non-sugar

**Table 3. Predicted mean per-item calories, saturated fat, trans fat, unsaturated fat, sugar, non-sugar carbohydrates, protein and sodium for newly introduced food and beverage items in 2013–2018.**

| Menu Category | n | New in 2013 | New in 2014 | New in 2015 | New in 2016 | New in 2017 | New in 2018 | p-value for trend | 2013–2018 | |
| --- | --- | --- | --- | --- | --- | --- | --- | --- | --- | --- |
| | | | | | | | | | Change | p-value |
| **Food**[a] | | | | | | | | | | |
| Calories (kcals) | 12308 | 618 | 602 | 585 | 588 | 525 | 547 | 0.10 | -71 kcals | 0.16 |
| Saturated fat (g) | 11934 | 10.8 | 11.0 | 10.3 | 10.4 | 10.6 | 9.6 | 0.38 | -1.2 g | 0.26 |
| Trans fat (g) | 10822 | 0.4 | 0.5 | 0.4 | 0.3 | 0.3 | 0.3 | 0.27 | -0.02 g | 0.84 |
| Unsaturated fat (g) | 10809 | 20.5 | 19.4 | 19.1 | 19.9 | 16.6 | 18.3 | 0.25 | -2.3 g | 0.30 |
| Sugar (g) | **10947** | **19.1** | **17.2** | **14.7** | **14.8** | **13.1** | **12.4** | **0.01** | **-6.7 g** | **0.02** |
| Non-sugar carbohydrates (g) | 10924 | 42.3 | 38.6 | 39.5 | 37.2 | 64.8 | 37.6 | 0.62 | -4.7 g | 0.27 |
| Protein (g) | 12068 | 26.6 | 26.1 | 24.9 | 25.6 | 22.9 | 25.9 | 0.51 | -0.7 g | 0.81 |
| Sodium (mg) | 12124 | 1315 | 1259 | 1228 | 1258 | 1113 | 1272 | 0.51 | -43 mg | 0.76 |
| **Beverages** | | | | | | | | | | |
| Calories (kcals) | 11646 | 316 | 355 | 304 | 283 | 300 | 231 | 0.23 | -85 kcals | 0.05 |
| Saturated fat (g) | 11027 | 5.0 | 6.0 | 5.2 | 3.7 | 3.8 | 1.6 | 0.20 | -3.4 g | 0.05 |
| Trans fat (g) | 10867 | 0.0 | 0.1 | 0.1 | 0.0 | 0.0 | 0.0 | 0.19 | -0.0 g | 0.26 |
| Unsaturated fat (g) | 10858 | 3.1 | 4.0 | 3.1 | 2.7 | 2.7 | 1.2 | 0.23 | **-1.8 g** | **0.01** |
| Sugar (g) | 11024 | 48.8 | 52.7 | 48.5 | 46.9 | 49.4 | 45.8 | 0.57 | -3.0 g | 0.63 |
| Non-sugar carbohydrates (g) | 10997 | 5.5 | 8.1 | 5.1 | 5.0 | 5.1 | 2.6 | 0.16 | **-2.9 g** | **0.01** |
| Protein (g) | **11403** | **6.1** | **5.9** | **5.4** | **4.9** | **4.8** | **3.4** | **<0.01** | **-2.7 g** | **<0.01** |
| Sodium (mg) | 11562 | 79 | 160 | 144 | 130 | 173 | 113 | 0.53 | 34 mg | 0.44 |

Boldface indicates statistical significance at *p*<0.05. The n indicates total number of items introduced in all years for that category. All estimates are adjusted for restaurant type, whether the restaurant is a national chain, the year the restaurant began labeling their menus with calories, and whether the item is categorized as a kid's item, shareable, regional or offered for a limited time.

[a] Included all menu categories except beverages and toppings & ingredients.

carbohydrates (-2.9g, -53%, p = 0.01, p-for-trend = 0.16) were significantly lower in 2018 compared to 2013 but the linear trend was not significant, indicating that the declines were not consistent across the study period. Within sub-categories of newly introduced items, we observed several changes (**S6 Table**). For example, calories decreased among burgers (-229 kcals, -23%, p = 0.01, p-for-trend = 0.02), while calories among sandwiches (-71 kcals, -11%, p = 0.04, p-for-trend = 0.23), desserts & baked goods (-111 kcals, -20%, p = 0.08, p-for-trend = 0.04), and salads (+113 kcals, 26%, p = 0.03, p-for-trend = 0.31), were significantly different in 2018 compared to 2013, with no significant linear trends appearing across study years.

## Sensitivity analyses

When we calorie-adjusted the nutrient analysis for newly introduced items (**S7 Table**), the significant declines in unsaturated fat (-0.6g, -21%, p = 0.02, p-for-trend = 0.22) and protein remained for beverages (-1.5g, -25%; p<0.01, p-for-trend = 0.01) while sugar (+9.6g, 21%, p = 0.02, p-for-trend = 0.09) and sodium (+73mg, 102%, p = 0.03, p-for-trend = 0.10) increased from 2013 to 2018, although the trend was not significant. The results did not change meaningfully for any of the further sensitivity analyses: 1) excluding items ±50 calories different than the sum of their nutrient calories or for which calories cannot be calculated due to

missing nutrients (**S8 Table**), 2) excluding items from restaurants that did not label their menus with calories by 2018 (**S9 Table**), and 3) reclassifying high-calorie items as shareable (**S10 Table**).

## Discussion

This study examined changes in calories and nutrients for more than 28,000 menu items at 66 of the largest U.S. chain restaurants from 2012 to 2018. Consistent with our hypothesis, we find that newly introduced menu items in large chain restaurants have continued to decline in calories through 2018 (a decrease which was primarily concentrated in the burgers and desserts/baked goods categories). Other changes in macronutrient content were sporadic and not clearly toward improved dietary quality. For example, we observed a significant increase in saturated fat from 2013 to 2018 but not a significant trend for that nutrient over the study period, and this result attenuated when we controlled for item calories. We observed no overall change in calories or nutrients for common items that were on the menu year-over-year.

The observed declines in calories and underlying changes in nutrients for newly introduced menu items are mostly consistent with prior studies;[9, 10, 14, 15] although our finding that average per-item sodium content increased for beverage items is different from an earlier paper showing a decrease in sodium for food items.[11] Like earlier studies, we found that the calories and nutrients of common items were not changing;[9, 10, 14] although one study did show a significant decline in calories in common items from 2008 to 2015.[9]

It is possible that the declines in calories and nutrients in this study are related to local or national nutrition policies; although this is unknowable from our data as we lack both pre and post implementation data for relevant policies. For example, the menu labeling rule included in the 2010 Affordable Care Act [19], was only implemented nationally on May 2018 (after several delays); the law mandates that calorie information be posted on menus and menu boards in restaurants or similar retail food establishments with more than 20 outlets. The data available for this study ends prior to the menu labeling implementation date. Prior to the Affordable Care Act, menu labeling regulations were adopted by more than 20 localities, beginning with New York City in 2008.[20] Another relevant policy is the National Salt Reduction Initiative (NSRI), which launched in 2009, a partnership with state and local health authorities and national health organizations that encouraged food manufacturers and restaurants to voluntarily reduce sodium in their products.[21, 22] A third policy is the ban on trans fats in U.S. restaurants and grocery stores which was announced by the Food and Drug Administration in 2015 (giving food-makers three years to phase out the ingredient).[23] Given that roughly a third of Americans eat at fast food restaurants on a typical day, this continued trend for newly introduced items may have implications for obesity and population health.[24] Without access to sales data, we do not know whether customers are ordering newly introduced menu items more frequently than common items, but the declining calorie content of new items provides customers with more opportunity to select lower calorie items. The observed reductions in calorie content of menu items–an average of 120 fewer calories in newly introduced items from 2013 to 2018 –may help reduce the excess calorie intake that underlie the obesity epidemic. Research suggests that small behavior changes that shift energy balance by 100 to 200 calories per day may be helpful for weight management.[25] While this energy deficit is smaller than current recommendations to produce clinical relevant weight loss [26], small calorie changes may be more sustainable than larger ones.[27] Moreover, these calories changes are likely unknown by consumers and such "stealth health" approaches may increase the possibility for sustained impact on calorie intake as they do not directly rely on consumer behavior changes. Moreover, because consumption of fast food is higher among racial and ethnic minority

populations, [24] and these groups are also at higher risk for obesity,[28] these changes may also help to promote improvements in health equity.

Possible gains to population health due to calorie declines in newly introduced items should be tempered by our finding that the overall nutritional quality of foods did not clearly change for the better. We observed declines in unsaturated fat and protein for all newly introduced items, sugar (for newly introduced food items), and protein (for newly introduced beverages). Not all of these changes are positive. The declines in unsaturated fat and protein may have implications for satiety.[29] Even though we did not empirically test for differences between common and newly introduced items, average calories are quantitatively lower in the most recent year of data collection. Time will tell if this remains. Overall, sugar and sodium increased in calorie-adjusted analyses, but not with a significant trend over the entire course of follow-up time.

Slight differences between the results in this manuscript and previous publications examining trends using the MenuStat data should be noted. First, we use the most current MenuStat data for this evaluation (obtained in June 2019), which includes updated data for previous years (confirmed by MenuStat via personal communication). Second, for our examination of calorie and nutrient trends among newly introduced menu items, we begin our comparison in 2013 whereas previous manuscripts have examined trends among new items compared to 2012.[9, 14] We used 2013 as the starting point because we could not determine whether items in 2012 were only on the menu in that year or also in prior years.

Future research should continue to monitor nutrient trends in large chain restaurants, particularly in response to the implementation of federal labeling policies.[19, 30] Several cities have introduced sodium labeling as well (i.e., New York City, Philadelphia), and it will be interesting to determine the effect of those labeling programs. Moreover, future work should examine the cause of calorie or nutrient decreases in menu items (e.g., decreases in volume, reformulation). It will also be important to collect and include data on purchases and consumption to determine if the observed changes in restaurant offerings are contributing to meaningful changes in purchasing and if these changes are realized by the populations most likely to consume restaurant food on a regular basis.[24, 31]

This study has several limitations. First, data are limited to menu items from 66 of the largest 100 U.S. chain restaurants and cannot be generalized to independent restaurants, small chains, or fine dining restaurants. Second, these data were collected from online menus posted on restaurant websites and may be subject to misreporting by restaurants or to human error in data entry and coding. Prior research has found that the stated calorie content of restaurant foods is generally accurate. [32]. However, MenuStat methods for data collection and entry are rigorous,[13] and results presented here remained robust after conducting several sensitivity analyses. Third, data for the MenuStat project are collected annually, in the first month of the calendar year. Therefore, items that are introduced between February and December of each year were attributed to the following year of data if they remain on the menu past the end of the calendar year. Fourth, while these data are a census of items available on restaurant menus, this analysis did not include price, purchase, or consumption data. Because of this, we cannot account for the popularity of menu items, the frequency of their orders, or the relative impact that this information may have on nutrient consumption among the population. Fifth, we had to remove 21% of menu items for which unable to determine whether they were common as we lack data before 2012 to know if they appeared before. Sixth, because we lack complete and consistent data on volume across all the MenuStat data, we are unable to examine changes in portion size over time. Seventh, because we lack sales and consumption data, we do not know which menu items customers are more likely to purchase; therefore, interferences about the impact of the study findings on consumer behavior is limited. Finally, this study did not

examine nutrient differences between restaurants that did or did not implemented calorie labeling. Previous research has shown that restaurants that post nutrition information have fewer per-item calories on their menus. [16] However, we did not have enough years of post-labeling information to fully assess any effects. Changes in nutrition after labeling has been implemented is an important area for future study, now that labeling was rolled out in May 2018.

From 2013 to 2018, newly introduced menu items in large chain restaurants declined in calories, which may help to reduce calorie intake. However, the macronutrient composition of newly introduced menu items did not shift to a healthier profile beyond calories. These findings suggest a continuing trend toward reducing calories in new menu items but little progress toward improving overall nutritional offerings at chain restaurants.

## Supporting information

**S1 File. Covariate definitions.**
(DOCX)

**S1 Table. Characteristics of 66 restaurants included in study.**
(DOCX)

**S2 Table. Restaurant- and item-level characteristics of items on menus from 2012–2018, overall and by category.**
(DOCX)

**S3 Table. Items missing nutrients on menus in 66 U.S. restaurants from 2012–2018, overall and by category.**
(DOCX)

**S4 Table. Predicted mean per-item calories, saturated fat, trans fat, unsaturated fat, sugar, non-sugar carbohydrates, protein and sodium for common items and for newly introduced items.**
(DOCX)

**S5 Table. Predicted mean per-item calories, saturated fat, trans fat, unsaturated fat, sugar, non-sugar carbohydrates, protein and sodium by category for common items, 2012–2018.**
(DOCX)

**S6 Table. Predicted mean per-item calories, saturated fat, trans fat, unsaturated fat, sugar, non-sugar carbohydrates, protein and sodium by category for newly introduced items, 2013–2018.**
(DOCX)

**S7 Table. Predicted calorie-adjusted per-item mean saturated fat, trans fat, unsaturated fat, sugar, non-sugar carbohydrates, protein and sodium for newly introduced items, 2013–2018.**
(DOCX)

**S8 Table. Predicted mean per-item calories, saturated fat, trans fat, unsaturated fat, sugar, non-sugar carbohydrates, protein and sodium for newly introduced items, 2013–2018.** Excluding items whose calories are ±50 calories different than the sum of their nutrient calories (N = 1149) or calories cannot be calculated due to missing nutrients (N = 559).
(DOCX)

**S9 Table. Predicted mean per-item calories, saturated fat, trans fat, unsaturated fat, sugar, non-sugar carbohydrates, protein and sodium for newly introduced items in 2013–2018.**

Excluding items from restaurants that did not label their menus with calories by 2018
(N = 338).
(DOCX)

**S10 Table. Predicted mean per-item calories, saturated fat, trans fat, unsaturated fat, sugar, non-sugar carbohydrates, protein and sodium for newly introduced items in 2013–2018 where all items >2000 calories (N = 24) and all appetizers & sides >1000 calories (N = 52) are flagged as shareable.**
(DOCX)

## Acknowledgments

The authors would like to thank the New York City Department of Health and Mental Hygiene for creating and maintaining Menustat.org.

## Author Contributions

**Conceptualization:** Sara N. Bleich.

**Formal analysis:** Mark J. Soto.

**Investigation:** Sara N. Bleich.

**Methodology:** Mark J. Soto, Caroline Glagola Dunn, Alyssa J. Moran, Jason P. Block.

**Project administration:** Caroline Glagola Dunn.

**Resources:** Sara N. Bleich.

**Supervision:** Sara N. Bleich.

**Writing – original draft:** Sara N. Bleich.

**Writing – review & editing:** Mark J. Soto, Caroline Glagola Dunn, Alyssa J. Moran, Jason P. Block.

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
