## [Decision Letter · Decision Letter 0]

27 Nov 2019

PONE-D-19-29367

Calorie and nutrient trends in large U.S. chain restaurants, 2012-2018

PLOS ONE

Dear Dr Bleich,

Thank you for submitting your manuscript to PLOS ONE. After careful consideration, we feel that it has merit but does not fully meet PLOS ONE’s publication criteria as it currently stands. Therefore, we invite you to submit a revised version of the manuscript that addresses the points raised during the review process.

We would appreciate receiving your revised manuscript by Jan 11 2020 11:59PM. To enhance the reproducibility of your results, we recommend that if applicable you deposit your laboratory protocols in protocols.io, where a protocol can be assigned its own identifier (DOI) such that it can be cited independently in the future. For instructions see: http://journals.plos.org/plosone/s/submission-guidelines#loc-laboratory-protocols

We look forward to receiving your revised manuscript.

Kind regards,

David Meyre

Academic Editor

PLOS ONE

Reviewers' comments:

Reviewer's Responses to Questions

**Comments to the Author**

1. Is the manuscript technically sound, and do the data support the conclusions?

Reviewer #1: Yes

Reviewer #2: No

Reviewer #3: Yes

2. Has the statistical analysis been performed appropriately and rigorously? 

Reviewer #1: Yes

Reviewer #2: No

Reviewer #3: Yes

3. Have the authors made all data underlying the findings in their manuscript fully available?

Reviewer #1: Yes

Reviewer #2: Yes

Reviewer #3: Yes

4. Is the manuscript presented in an intelligible fashion and written in standard English?

Reviewer #1: Yes

Reviewer #2: Yes

Reviewer #3: Yes

5. Review Comments to the Author

Reviewer #1: This paper is on a timely subject and addresses the question of how the calories and nutritional content of menu items are changing over time. Though I’m overall enthusiastic about this paper, there are several concerns that I believe should be addressed before this paper is suitable for publication. I have two major concerns:

1) The subgroup analyses looking at the sub-types of food seems like an afterthought whereas to me it is of far greater relevance than many of the sensitivity analyses conducted.

2) In Table 1 and other similarly laid out tables, the percentages presented do not appear to make sense. Please check the percentages and if correct, revise titles, headings, or include footnotes to make it clear what the numerator and denominator is for these calculations.

Beyond these major concerns, there are many opportunities to provide clarity for what is a large and complicated paper. Additionally, the discussion would benefit from more caution about extrapolating changes in menus to consumer behavior.

Specific concerns:

Line 21: The objectives of your analyses should be more clear in your abstract.

Line 43: Change to: “Compared to eating at home, eating in restaurants is associated with consuming more calories…”.

Line 45: Move to after the first sentence of this paragraph discussing costs of eating away from home.

Line 48: The timeline explaining which years of menu items are being compared to difficult to following. Please clarify.

Line 68: Please elaborate on which restaurants are included in the Menustat Project and how they are selected. In line 57 you said that the majority of the Top 100 restaurants are included, but why not all?

Line 77: Which nutrients are you specifically interested in?

Line 85: I find the description of the models difficult to interpret. Consider dividing how you conducted the analyses for the common items versus the newly introduced item to improve clarity.

Line 88: In which situations was year a continuous variable versus in which was it a categorical variables?

Line 99: In your main analyses, what did you not adjust for the sub-type of food?

Line 105: In your discussion, the elimination of 21% of the food items that were on a menu in 2012 and remove din any year prior to 2018 should be mentioned as a serious limitation. Could you do a sensitivity analyses where these items are included and are considered a new item as of 2012?

Line 121: Please elaborate on what you mean be “we accounted for clustered observations at the restaurant chain level because items within chains may have been correlated”. More broadly, this seems to be referring to the model building process. I would consider restructuring the methods section to include the details on model building currently found in lines 85 to 99 to fit under the statistical analyses section.

Line 128: None of these sensitivity analyses are as important as your subgroup analyses looking at the sub-types of food. I would consider just including this in the discussion (as you already have) as a simple “the results were robust when we did x, y, z, look at the Supplementary Appendices). Instead, you should write about how you looked at individual food categories and the table related to this should be part of the main article.

Results in general: Please add subheadings. You have conducted a lot of analyses and it is difficult to follow the results as written. Subheadings will help orient your readers.

Figure 1: What does the cross mean?

Table 1 – I can’t figure out what the percentages are referring to in this table. For example for appetizers and sides, the n (%) column is 1722 and 6.1% which is adding up all the items and percentages from each year and the items common to all years. However, to calculate the percentage, the denominator should be all food items? I also have no idea what the percentages for each of the columns within each category are referring to. Please clarify and check all tables to make sure this isn’t a recurring issue.

Line 136: It seems odd that you have chosen to do a subgroup analyses within coffee chains only based on the findings of one regional study. Particularly considering that you did no other stratification by restaurant type which seems like a far more relevant variable to consider. Either better justify this analyses or consider excluding it. Furthermore, the connection between evidence showing that people selected different foods because of menu labeling and that leading to changes in menu items requires better justification.

Line 193: The subgroup analyses is missing from elsewhere in this paper. Generally, it seems inappropriate to pool data from all the different sub-categories of food items. Based on Table 1, it appears that there were changes in what percentage of menu items belonged to each category by year which would impact the results. I would make supplementary table 5 a main table in your paper.

Line 228: Why would you think that the findings would be the same for food versus beverages?

Line 239: I think it’s a stretch to pull in health equity and minority populations and which specific minority populations are you talking about? Ethnicity, sexual orientation, religion? Given that this paragraph is saying that you don’t have sales data and therefore don’t know what people are ordering, you have no idea if minority populations really have a better opportunity for health because of the changes observed in new menu item calorie content.

Line 233 – 241: This section would benefit from the addition of discussing if the changes in calories is really meaningful from a clinical stand point. With such a large sample size, it’s easy to find statistically significant results, but they don’t always translate over to a meaningful difference.

Line 251: What is the implication of limited time items and other “specials” that are being captured as newly introduced items?

Line 251: Decreased calories purchased in coffee chains based on menu labeling doesn’t mean that the newly introduced items aren’t going to have more calories.

Line 253: To verify the statement that “this suggests that customers looking to make lower calorie purchases in this setting may have increasingly fewer options” is actually true, you need to connect to a result from your paper. If the common items aren’t changing nutritionally and these “new” items are perhaps replacing old items, isn’t there the potential that the caloric content of the options available to people is actually more static? Furthermore, your results clearly showed that the calorie content of newly introduced items is decreasing over time, therefore it seems like restaurants are trying to make lower calorie options more readily available. If you’re going to make this statement, you need to back it up with substantial evidence.

Line 255: I would suggest either removing all this emphasis on coffee chains, or instead providing a very clear explanation of why they are so uniquely different from the other categories of restaurants that they deserve specific attention.

Line 269: Decreases in what?

Line 276: Somewhere earlier in your methods you should be mentioning what types of restaurants are included – why does the database exclude fine dining restaurants? This is information that helps contextualize the results and is important to be upfront with.

Line 277: In addition to restaurants misreporting and human error, what sort of precision is there when nutritional estimates are made on restaurant items?

Line 289: Please provide a citation regarding menu labeling practices and calorie content of the menus.

Supplementary Table 5: I would be wary of how you interpret categories where the p-value for the trend is not significant but the p-value for change is. These seem to indicate when the items in 2013 or 2018 were unusually low/high in that specific nutrient.

Reviewer #2: This is a very informative paper describing changes in nutritional content among “common” items that have been on the chain restaurant menus for at least 6 years and those that are newly introduced. In order to understand what these data mean, it would be helpful if the authors elaborated more on the background and context. For example, how many restaurants are not covered by these 66 chain restaurants? (ie non-chain restaurants) How do the chain restaurants that were studied differ from the 34 chain restaurants not chosen? Understanding how these fit in the overall food environment could give us a more tangible sense of what the findings might mean for population health.

The analysis covered only common items and new items. It appears that common items comprise only 20% of food and 10% of beverage items, while the new items are 12-20% and 10-25% of food and beverages, respectively. You should make it clear that this refers to each year (if that’s the case). These don’t add up to 100%, or even 50% of all items, which means the bulk of the items on the restaurant menu are not being assessed at all. Why leave out the bulk of the menu?

Figure 1 does not match with Tables 2 and 3. For example, Calories in Figure 1 show the newly introduced calories are lower than in common items, but in table 2 it shows common items in 2019 were 467 and in Table 3 new items in 2018 had an average of 547 calorie. Is that a typo? Figure 1 shows about 350 calories. Same with saturated fat. Table 2 says 8.6 in 2018 and Table 3 has 9.6 in 2018, but the graph in Figure 1 shows < 5. What am I missing?

I found the presentation of nutrients in beverages confusing, as it is hard to think about beverages having unsaturated fat or protein, except for dairy type beverages. Do beverages have much added sodium? Maybe for unusual drinks like hot chocolate, vegetable drinks (tomato juice) or salted lassi—but aren’t most beverages very low in sodium? How can you explain an increase? How are the beverages combined? Do you think it is fair to combine sodas and milkshakes, for example? How are these combined? How do you account for variety? How do you account for size? Are large medium and small drinks averaged? If a menu has 10 sodas, 5 diet sodas and 1 milkshake option and another has 2 diet sodas and 10 milkshake options, the average calories will show a very different picture, and the result is a reflection of variety or portion size. This doesn’t seem a very meaningful analysis without going into more detail in the methods.

The trend in the new items in improved nutrient content is not of as much interest as the finding that their nutrient content is consistently worse than the common items, especially in terms of sugar. To me that’s a big story, that chain restaurant are putting more sugar in their new items, at a time when sugar is found to be culpable for many chronic diseases. It’s misleading to highlight in the discussion that new items have declines in calories, when they continue to be higher than the common ones. The conclusion should be revised, as it suggests that the new items may reduce calorie intake, when in fact if people consume them instead of the common items, they will be increasing their calorie intake.

Reviewer #3: This article is a relevant update of previous work on an important issue, the evolution of a major source of food in the US diet over time, and in relation to the national implementation of a long-delayed component of the Affordable Care Act, mandatory nationwide menu labeling. While menu labeling had spread widely even before the baseline year it became mandatory in 2018, which would suggest the possibility of a stronger incentive for change. Although the changes covered by the article , whic appear to end in January 2018, cover the roll-out only to 4 months before mandatory labeling.

While the article addresses only foods offered, not foods consumed, understanding to what extent menu labeling and /or changes in consumer demand have driven menu reformulation is an essential part of the puzzle of documenting the response of the food industry to the obesity epidemic and other nutritional challenges for noncommunicable disease prevention. For that reason this and other work documenting changes in the nutritional quality of foods, now a leading determinant of the burden of chronic disease, is important.

One weakness of the article was that it failed to clearly contextualize nutrition policy initiatives that were occurring at the same time - these included not only the dissemination of menu labeling which began in 2008 and became nationally mandatory in 2018, but also the National Salt Reduction Initiative and the FDAs voluntary targets for sodium reduction, and the prohibition of trans fats, also beginning in NYC in 2007 and spreading nationally by 2015. This timing should be more clearly laid out.

I was also unclear why the year of adoption of menu labeling was added to the modeling, and did not see the clear presentation of analysis of that variable in the adjusted model.

In general it would be helpful if the tables included not just the change in grams of each nutrient or calories but in %. This facilitates understanding whether the changes were significant not just statistically but practically. The decline in calories was 25% in new items , which is actually quite substantial.

In regards to interpretation of any increase in saturated fat, it would be helpful to also display trans fat and trans fat + saturated fat. The national ban on partially hydrogenated oils went into effect during the period of the study. While many chains had already gotten rid of artificial trans fat a quick perusal of Menustat suggests tat some were still using PHOs. Ruminant trans fat of course continues present. But if the goal was for trans fat to be replaced by healthier fats to the extent possible, ideally that total of Trans + saturated fats would decline even if saturated fat increased slightly, since saturated were needed to some extent to replace PHOs, particularly in baking applications

In limitations, the absence of sufficient information on portion size should also be noted, which would have helped to interpret whether these changes represent less calorie dense foods or just less food. This requirement is not part of mandatory nutrition information under ACA. As noted much of the change in nutrients reflected overall calorie content.

S1 Table - It would be helpful if the table also provided the year menu labeling was added by each company

6. PLOS authors have the option to publish the peer review history of their article (what does this mean?). If published, this will include your full peer review and any attached files.

Reviewer #1: Yes: Alexandra J Mayhew

Reviewer #2: No

Reviewer #3: No

---

## [Author Response · Author response to Decision Letter 0]

19 Jan 2020

Reviewer #1

1. This paper is on a timely subject and addresses the question of how the calories and nutritional content of menu items are changing over time. Though I’m overall enthusiastic about this paper, there are several concerns that I believe should be addressed before this paper is suitable for publication. I have two major concerns:

Response: Thank you for this feedback. Below we provide specific responses to your comments.

2. The subgroup analyses looking at the sub-types of food seems like an afterthought whereas to me it is of far greater relevance than many of the sensitivity analyses conducted.

Response: We agree that the subgroup analyses are an important aspect of the paper. We felt it was critical to assess this and did not intend for these analyses to come across as an afterthought. We actually included this as part of our a priori hypothesis (pertinent text bolded below):

“Building on prior research, we hypothesized that calories and nutrients among items commonly on the menu each year remained the same and that calories and nutrients most associated with chronic disease risk (sugar, saturated fat, sodium) decreased among newly introduced items, with some variability across menu categories.”

To emphasize this to the reader, we have also updated the methods (new text underlined below): 

“We controlled for the covariates described above, and the margins command was used to estimate the yearly predicted mean per-item nutrient values for each menu category and sub-category (e.g., all items; items categorized as food or beverage; food subcategorized as appetizer or side, main dish, etc.) post-regression.”

In the results section, we already include the below summary of the significant results for the subgroup analysis:

“Within sub-categories of newly introduced items, we observed several changes (S6 Table). For example, calories decreased among burgers (-229 kcals, -23%, p=0.01, p-for-trend=0.02), while calories among sandwiches (-71 kcals, -11%, p=0.04, p-for-trend=0.23), desserts & baked goods (-111 kcals, -20%, p=0.08, p-for-trend=0.04), and salads (+113 kcals, 26%, p=0.03, p-for-trend=0.31), were significantly different in 2018 compared to 2013, with no significant linear trends appearing across study years.”

Please let us know if there is more that we can do to highlight these important findings.

3. In Table 1 and other similarly laid out tables, the percentages presented do not appear to make sense. Please check the percentages and if correct, revise titles, headings, or include footnotes to make it clear what the numerator and denominator is for these calculations.

Response: Thanks for this comment. The percentages were correct in the original submission. However, they were presented in a confusing way since the super categories (MenuStat categories) and sub categories (main course categories) separately add to 100% but were not clearly delineated. Now, we present super and sub categories in separate sections for Table 1 (and Table S3 [formerly table S2]) and each sums to 100%. We hope these changes make the tables easier to understand.

4. Beyond these major concerns, there are many opportunities to provide clarity for what is a large and complicated paper. Additionally, the discussion would benefit from more caution about extrapolating changes in menus to consumer behavior. 

Response: Thank you for this feedback. Below we provide specific responses to your comments about clarification and changes to the discussion. Below is the text we added the limitations section regarding the implications for consumer behavior:

“Seventh, because we lack sales and consumption data, we do not know which menu items customers are more likely to purchase; therefore, interferences about the impact of the study findings on consumer behavior is limited.”

5. Line 21: The objectives of your analyses should be clearer in your abstract.

Response: We have revised the abstract. New text underlined:

“The objective of this study was to provide updated calorie trends through 2018 and examine trends in the macronutrient composition of menu items across this time period.”

6. Line 43: Change to: “Compared to eating at home, eating in restaurants is associated with consuming more calories…”.

Response: Done

7. Line 45: Move to after the first sentence of this paragraph discussing costs of eating away from home.

Response: Done

8. Line 48: The timeline explaining which years of menu items are being compared to difficult to following. Please clarify.

Response: We have revised this text as follows (new text underlined):

“In recent years, large chain restaurants have made calorie and nutrient changes to menu items which may benefit population health. From 2012 to 2014, the calorie content of newly introduced menu items decreased by about 60 calories (or 12 percent) [1]. From 2012 to 2015, mean calories for items that remained on the menu were significantly lower than mean calories for items that were dropped from the menu.[1] Among fast food restaurants, this decline was driven primarily by a decrease in calories from unsaturated fat.[2] From 2012 to 2016, calorie-adjusted sodium content in newly introduced menu items declined by 104mg.[3]”

9. Line 68: Please elaborate on which restaurants are included in the MenuStat Project and how they are selected. In line 57 you said that the majority of the Top 100 restaurants are included, but why not all?

Response: MenuStat contains nutrition data collected from the largest 100 restaurant chains in the country, by U.S. sales (Nation’s Restaurant News). Data on MenuStat are sourced from restaurant websites, and so only restaurants that post nutrition information online are available in MenuStat. When data collection for the MenuStat Project began in 2012, only 66 of the largest 100 restaurant chains posted nutrition information online. Since we are interested in looking at trends, this analysis focuses on those restaurants which have online nutrition data beginning in 2012. As a result, our sample size is 66 restaurants.

10. Line 77: Which nutrients are you specifically interested in?

Response: We are interested in sodium, sugar and saturated fat as they are nutrients of high public health importance. We have added the following text:

“We are particularly interested in changes in sodium, sugar and saturated fat, and display those results graphically, as they are nutrients of high public health importance.”

11. Line 85: I find the description of the models difficult to interpret. Consider dividing how you conducted the analyses for the common items versus the newly introduced item to improve clarity.

Response: We have clarified the language by clearly defining each of the two models we are using for the analysis in our “Measures” section: 

We examined two sets of continuous outcomes: 1) mean, within-item changes in calories and nutrients (e.g., sodium, sugar and saturated fat) from 2012 to 2018 among items on the menu in all years (hereafter referred to as common items) and 2) mean, between-item calories and nutrients among newly introduced items from 2013 to 2018. We are particularly interested in changes in sodium, sugar and saturated fat, and display those results graphically, as they are nutrients of high public health importance. Consistent with prior studies using these data,[1-6] menu items were identified as common if they had the same name, item description, and item ID in all years; items were defined as newly introduced in 2013-2018 if they did not have a matching name, item description, or item ID in any preceding year.

We have also moved some text to the “Statistical Analysis” section. Below are sections from the manuscript (new text underlined) that provide details about the two sets of models:

“For each set of outcomes (calories and nutrients in common items and newly introduced items), we examined two models. First, to test for a linear trend, the main independent variable was a continuous variable for the year the item was offered (common items) or introduced (newly introduced items). Second, to compare differences between 2012 vs. 2018 (common items) or 2013 vs. 2018 (newly introduced items), year was specified as a categorical variable. For both models, we included item-level covariates (children’s menu item status, regional, limited-time offers) and whether the item is categorized as shareable (also a MenuStat designation) which is similar to prior publications using the MenuStat data,[1-5, 7]. To account for correlated observations over time for common items only, we also included item fixed effects and removed all time-invariant restaurant-level covariates. For relevant models, we report either p-values for trend to indicate linear changes over time or p-values to indicate changes between the first and last year of the data.”

12. Line 88: In which situations was year a continuous variable versus in which was it a categorical variables?

Response: Year was specified as a continuous variable in models that examined for a linear trend across all years. We used year as a categorical variable when we compared differences between 2012 vs. 2018 (common items) or 2013 vs. 2018 (newly introduced items). This should be clearer now due to the revisions on the prior comment. 

13. Line 99: In your main analyses, what did you not adjust for the sub-type of food?

Response: We did not adjust for the specific MenuStat categories in the main analyses because we were interested in the availability of calories on the menu, which could be due to changes in the types of items available (e.g., more entrees, fewer sides) or due to changes in the caloric composition of items within categories.

14. Line 105: In your discussion, the elimination of 21% of the food items that were on a menu in 2012 and removed any year prior to 2018 should be mentioned as a serious limitation. Could you do a sensitivity analyses where these items are included and are considered a new item as of 2012?

Response: We have added the following text to the limitations section:

“Fifth, we had to remove 21% of menu items for which we were unable to determine whether they were common or newly introduced as we lack data before 2012 to know if they appeared before.”

With respect to a possible sensitivity analysis which includes the 21% of menu items, we made a study design decision to exclude these items because we lacked data prior to 2012 to determine when items were introduced that appeared in 2012. Pursuing the sensitivity analysis would require us to change our study design – a revision which we would prefer not to do. However, if the Editor feels strongly about this, we would happy to conduct this analysis.

15. Line 121: Please elaborate on what you mean be “we accounted for clustered observations at the restaurant chain level because items within chains may have been correlated”. More broadly, this seems to be referring to the model building process. I would consider restructuring the methods section to include the details on model building currently found in lines 85 to 99 to fit under the statistical analyses section.

Response: We have added the following clarifying text (new text underlined): 

“We accounted for clustered observations at the restaurant chain level because items within chains may have been correlated, meaning that menu items within a restaurant chain may be more similar than menu items between restaurant chains.”

We have also moved the description of the models to the Statistical Analysis section.

16. Line 128: None of these sensitivity analyses are as important as your subgroup analyses looking at the sub-types of food. I would consider just including this in the discussion (as you already have) as a simple “the results were robust when we did x, y, z, look at the Supplementary Appendices). Instead, you should write about how you looked at individual food categories and the table related to this should be part of the main article.

Response: We believe it is important to describe the statistical approach for the sensitivity analyses before providing readers with those results. We could certainly include the supplemental table, which includes the results for each MenuStat category, in the main text. However, we worry that the table is a bit too long and is better suited for the supplementary materials. 

17. Results in general: Please add subheadings. You have conducted a lot of analyses and it is difficult to follow the results as written. Subheadings will help orient your readers.

Response: Done

18. Figure 1: What does the cross mean?

Response: We have revised the note for the cross below the figure as follows:

 “The estimate for the change in calories from 2013 to 2018 is significant at p<0.05.”

19. Table 1 – I can’t figure out what the percentages are referring to in this table. For example for appetizers and sides, the n (%) column is 1722 and 6.1% which is adding up all the items and percentages from each year and the items common to all years. However, to calculate the percentage, the denominator should be all food items? I also have no idea what the percentages for each of the columns within each category are referring to. Please clarify and check all tables to make sure this isn’t a recurring issue.

Response: As mentioned above, the percentages were correct in the original submission. However, they were presented in a confusing way since the super categories (MenuStat categories) and sub categories (main course categories) separately add to 100% but were not clearly delineated. Now, we present super and sub categories in separate sections for Table 1 (and Table S3 [formerly table S2]) and each sums to 100%. We hope these changes make the tables easier to understand.

20. Line 136: It seems odd that you have chosen to do a subgroup analyses within coffee chains only based on the findings of one regional study. Particularly considering that you did no other stratification by restaurant type which seems like a far more relevant variable to consider. Either better justify this analyses or consider excluding it. Furthermore, the connection between evidence showing that people selected different foods because of menu labeling and that leading to changes in menu items requires better justification.

Response: We have excluded the coffee chain sub-analysis from the manuscript. To the point about consumer behavior, we have added the following caveat to the limitations section:

“Seventh, because we lack sales and consumption data, we do not know which menu items customers are more likely to purchase; therefore, interferences about the impact of the study findings on consumer behavior is limited.”

21. Line 193: The subgroup analyses is missing from elsewhere in this paper. Generally, it seems inappropriate to pool data from all the different sub-categories of food items. Based on Table 1, it appears that there were changes in what percentage of menu items belonged to each category by year which would impact the results. I would make supplementary table 5 a main table in your paper.

Response: Supplemental table 5 is now supplemental table 6. As mentioned above, we worry that this table may be too large to add to the main document. Our preference would be to leave it in the supplemental materials for an interested reader. That said, if the editor feels strongly about this, we are happy to make the change. 

22. Line 228: Why would you think that the findings would be the same for food versus beverages?

Response: If retailers are making changes to improve the health content of items, then that should affect both food and beverages. We found some diverging evidence for this with some rise in sodium content of newly introduced beverages but not food; this was found only in calorie-adjusted analyses. This change might have occurred in beverages but not food because beverage availability in large chain restaurants has substantially increased in quantity and variety over the study period (Frelier, AJPM, 2019).

23. Line 239: I think it’s a stretch to pull in health equity and minority populations and which specific minority populations are you talking about? Ethnicity, sexual orientation, religion? Given that this paragraph is saying that you don’t have sales data and therefore don’t know what people are ordering, you have no idea if minority populations really have a better opportunity for health because of the changes observed in new menu item calorie content.

Response: We agree that we cannot state whether choices are differing by subgroups of the population, because this study is focused on what is available on the menu. However, we know that consumption of fast food is higher among racial and ethnic minority populations, and that these groups are also at higher risk for obesity. If restaurants are changing to offerings that are lower calorie, this could provide an opportunity for healthier choices among frequent consumers of restaurant food. Your point is well taken that we have only focused, in our comments, on equity based on race/ethnicity. To address your point about our general use of the term “minority,” we have edited this section to refer specifically to racial and/or ethnic minorities, as has been documented in the literature. 

24. Line 233 – 241: This section would benefit from the addition of discussing if the changes in calories is really meaningful from a clinical stand point. With such a large sample size, it’s easy to find statistically significant results, but they don’t always translate over to a meaningful difference.

Response: Good suggestion. We agree that the differences are small, but they are still relevant. We have added the following text to put these absolute differences in context:

“Research suggests that small behavior changes that shift energy balance by 100 to 200 calories per day may be helpful for weight management.[9] While this is amount of energy deficit is smaller than current recommendations to produce clinical relevant weight loss,[10] small calorie changes may be more sustainable than larger ones.[11] Moreover, these calories changes are likely unknown by consumers and such “stealth health” approaches may increase the possibility for sustained impact on calorie intake as they do not directly rely on consumer behavior changes.”

25. Line 251: What is the implication of limited time items and other “specials” that are being captured as newly introduced items?

Response: Limited time items make up a small percentage of MenuStat data (~5% of data), so we do not expect these items to substantially impact the results for newly introduced menu items. Our models do adjust for whether an item is offered for a limited time. If we drop these items from the analysis, our results do not substantially change (see table below). If you believe this table should be in the manuscript, we can add it as a supplement. 

Response Table 1. Predicted mean per-item calories, saturated fat, unsaturated fat, sugar, non-sugar carbohydrates, protein and sodium for newly introduced, non-limited time offer food and beverage items in 2013-2018.

Menu Category n New in 2013 New in 2014 New in 2015 New in 2016 New in 2017 New in 2018 p-value for trend 2013-2018

 Change p-value

Fooda 

Calories (kcals) 11633 622 599 582 589 528 545 0.10 -77 kcal 0.14

Saturated fat (g) 11261 10.7 11.1 10.1 10.4 10.6 9.5 0.39 -1.2 g 0.24

Trans fat (g) 10195 0.4 0.5 0.4 0.3 0.3 0.3 0.33 -0.0 g 0.89

Unsaturated fat (g) 10182 20.7 19.3 19.0 20.1 16.7 18.4 0.27 -2.3 g 0.30

Sugar (g) 10323 18.9 17.3 14.4 14.7 12.9 11.8 0.01 -7.2 g 0.02

Non-sugar carbohydrates (g) 10300 42.9 37.7 40.1 37.3 67.3 37.4 0.62 -5.5 g 0.21

Protein (g) 11397 26.9 25.9 24.8 25.7 23.1 26.0 0.53 -0.9 g 0.75

Sodium (mg) 11449 1334 1233 1234 1265 1125 1289 0.60 -45 mg 0.76

Beverages 

Calories (kcals) 10851 306 286 291 266 311 218 0.25 -89 kcal 0.01

Saturated fat (g) 10238 4.7 3.5 4.8 3.2 4.2 1.7 0.21 -3.0 g 0.02

Trans fat (g) 10082 0.0 0.1 0.1 0.0 0.0 0.0 0.20 -0.0 g 0.05

Unsaturated fat (g) 10073 2.9 2.1 2.9 2.3 3.0 1.3 0.29 -1.6 g <0.01

Sugar (g) 10238 48.0 47.9 47.1 45.1 50.8 42.6 0.76 -5.3 g 0.36

Non-sugar carbohydrates (g) 10211 5.2 5.7 4.7 4.5 5.5 2.7 0.13 -2.4 g <0.01

Protein (g) 10610 5.9 5.0 5.1 4.6 4.6 3.2 <0.01 -2.7 g <0.01

Sodium (mg) 10768 75 105 137 121 180 116 0.17 42 mg 0.25

Note. Boldface indicates statistical significance at p<0.05. The n indicates total number of items introduced in all years for that category. All estimates are adjusted for restaurant type, whether the restaurant is a national chain, the year the restaurant began labeling their menus with calories, the item’s food category and whether the item is categorized as a kid’s item, shareable, regional or offered for a limited time.

a Included all menu categories except beverages and toppings & ingredients.

26. Line 251: Decreased calories purchased in coffee chains based on menu labeling doesn’t mean that the newly introduced items aren’t going to have more calories.

Response: We agree, but we thought it was important to make this point to readers since – as we mention in the text – customers looking to make lower calorie purchases in this setting may have increasingly fewer options as calories in newly introduced beverages in coffee chains have mostly increased. However, in this revised version, we have removed the coffee chain sub analysis from the paper.

27. Line 253: To verify the statement that “this suggests that customers looking to make lower calorie purchases in this setting may have increasingly fewer options” is actually true, you need to connect to a result from your paper. If the common items aren’t changing nutritionally and these “new” items are perhaps replacing old items, isn’t there the potential that the caloric content of the options available to people is actually more static? Furthermore, your results clearly showed that the calorie content of newly introduced items is decreasing over time, therefore it seems like restaurants are trying to make lower calorie options more readily available. If you’re going to make this statement, you need to back it up with substantial evidence.

Response: Thanks for this comment. The text you flagged is only relevant to coffee chains, and this sensitivity analysis has been removed from the manuscript. We do know from other analyses (Bleich et al, AJPM 2017) that items dropped from chain restaurant menus are significantly higher in calories than items that remain on the menu. 

28. Line 255: I would suggest either removing all this emphasis on coffee chains, or instead providing a very clear explanation of why they are so uniquely different from the other categories of restaurants that they deserve specific attention.

Response: As mentioned above, we have excluded the coffee chain sub-analysis from the manuscript. 

29. Line 269: Decreases in what?

Response: This is referring to calorie or nutrient decreases. We have updated the text.

30. Line 276: Somewhere earlier in your methods you should be mentioning what types of restaurants are included – why does the database exclude fine dining restaurants? This is information that helps contextualize the results and is important to be upfront with.

Response: We have added the following sentence at the start of the methods (new text underlined):

“We restricted our analysis to 66 fast food, fast casual, and full service restaurant chains that appear in all seven years of data (2012-2018), which includes N=28,238 unique menu items (see S1 Table for details).”

In terms of why fine dining restaurants are excluded, we are limited to the restaurants included in the MenuStat database. As mentioned above, MenuStat contains nutrition data collected from the largest 100 restaurant chains in the country, by U.S. sales (Nation’s Restaurant News). 

31. Line 277: In addition to restaurants misreporting and human error, what sort of precision is there when nutritional estimates are made on restaurant items?

Response: Prior research suggests that Calorie estimates of restaurant food are accurate overall. We have modified this sentence (new text underlined):

“Second, these data were collected from online menus posted on restaurant websites and may be subject to misreporting by restaurants or to human error in data entry and coding. Prior research has found that the stated calorie content of restaurant foods is generally accurate [12]. ”

32. Line 289: Please provide a citation regarding menu labeling practices and calorie content of the menus

Response: Done

33. Supplementary Table 5: I would be wary of how you interpret categories where the p-value for the trend is not significant but the p-value for change is. These seem to indicate when the items in 2013 or 2018 were unusually low/high in that specific nutrient.

Response: We agree. In the discussion where we talk about the results from Supplementary Table 6, formerly Table S5 (which presents the results for each MenuStat category) we now identify findings in which we observed significant changes between 2013 vs 2018 compared to when we saw significant changes for the linear trend. 

Reviewer #2

1. This is a very informative paper describing changes in nutritional content among “common” items that have been on the chain restaurant menus for at least 6 years and those that are newly introduced. In order to understand what these data mean, it would be helpful if the authors elaborated more on the background and context. For example, how many restaurants are not covered by these 66 chain restaurants? (ie non-chain restaurants) How do the chain restaurants that were studied differ from the 34 chain restaurants not chosen? Understanding how these fit in the overall food environment could give us a more tangible sense of what the findings might mean for population health.

Response: Good point. To give readers a sense of how the restaurants included in this study fit within the overall food environment, we have added the following text to the introduction (new text underlined):

“MenuStat is a database that tracks nutritional content of menu items, currently through 2018, in nearly all of the 100 top-selling U.S restaurant chains, representing about half of all US restaurant revenue [13].”

Unfortunately, we lack data on how the restaurants included in the study differ from those which are not included – this point is already made in the limitations section. 

2. The analysis covered only common items and new items. It appears that common items comprise only 20% of food and 10% of beverage items, while the new items are 12-20% and 10-25% of food and beverages, respectively. You should make it clear that this refers to each year (if that’s the case). These don’t add up to 100%, or even 50% of all items, which means the bulk of the items on the restaurant menu are not being assessed at all. Why leave out the bulk of the menu?

Response: Thanks for this comment. This was also flagged by the Reviewer #1. The percentages were presented in a confusing way in the original submission since the super categories (MenuStat categories) and sub categories (main course categories) separately add to 100% but were not clearly delineated. Now, we present super and sub categories in separate sections for Table 1 (and Table S3 [formerly table S2]) and each sums to 100%. We hope these changes make the tables easier to understand.

3. Figure 1 does not match with Tables 2 and 3. For example, Calories in Figure 1 show the newly introduced calories are lower than in common items, but in table 2 it shows common items in 2019 were 467 and in Table 3 new items in 2018 had an average of 547 calorie. Is that a typo? Figure 1 shows about 350 calories. Same with saturated fat. Table 2 says 8.6 in 2018 and Table 3 has 9.6 in 2018, but the graph in Figure 1 shows < 5. What am I missing?

Response: Yes, that’s correct. Fig 1 shows predicted means among common and newly introduced items (combining food and beverages together) whereas Tables 2 and 3 separate food and beverages. We did this intentionally for two reasons: 1) to provide readers with the overall trends as well as the break down for food and beverages and 2) to ensure that the information was not redundant between the tables and the figures. 

4. I found the presentation of nutrients in beverages confusing, as it is hard to think about beverages having unsaturated fat or protein, except for dairy type beverages. Do beverages have much added sodium? Maybe for unusual drinks like hot chocolate, vegetable drinks (tomato juice) or salted lassi—but aren’t most beverages very low in sodium? How can you explain an increase? How are the beverages combined? Do you think it is fair to combine sodas and milkshakes, for example? How are these combined? How do you account for variety? How do you account for size? Are large medium and small drinks averaged? If a menu has 10 sodas, 5 diet sodas and 1 milkshake option and another has 2 diet sodas and 10 milkshake options, the average calories will show a very different picture, and the result is a reflection of variety or portion size. This doesn’t seem a very meaningful analysis without going into more detail in the methods.

Response: In a separate paper by our group (Frelier, AJPM, 2019), which we reference in our current submission, we used the same data and to look at trends in calorie and nutrient content of beverages and examined a number of beverage sub-categories. So, we intentionally did not break out beverages into sub-categories for this analysis, instead pointing our reader to this previous publication for more detailed analysis. We believe there is value in looking at beverages overall (as we do in this paper to understand overall trends) and among specific sub-categories (as we did in the other paper). We do not account for size in this analysis, so the reviewer is correct that average calories could reflect ounces and/or beverage type. 

5. The trend in the new items in improved nutrient content is not of as much interest as the finding that their nutrient content is consistently worse than the common items, especially in terms of sugar. To me that’s a big story, that chain restaurant are putting more sugar in their new items, at a time when sugar is found to be culpable for many chronic diseases. It’s misleading to highlight in the discussion that new items have declines in calories, when they continue to be higher than the common ones. The conclusion should be revised, as it suggests that the new items may reduce calorie intake, when in fact if people consume them instead of the common items, they will be increasing their calorie intake.

Response: In terms of calories, it is notable that average calories in newly introduced items are lower than common items in the most recent year of the data. This is illustrated in Figure 1. The point about sugar is a good one. We have added the following text to the discussion:

“Even though we did not empirically test for differences between common and newly introduced items, average calories are qualitatively lower in the most recent year of data collection. Time will tell if this remains.”

Reviewer #3

1. This article is a relevant update of previous work on an important issue, the evolution of a major source of food in the US diet over time, and in relation to the national implementation of a long-delayed component of the Affordable Care Act, mandatory nationwide menu labeling. While menu labeling had spread widely even before the baseline year it became mandatory in 2018, which would suggest the possibility of a stronger incentive for change. Although the changes covered by the article, which appear to end in January 2018, cover the roll-out only to 4 months before mandatory labeling.

While the article addresses only foods offered, not foods consumed, understanding to what extent menu labeling and /or changes in consumer demand have driven menu reformulation is an essential part of the puzzle of documenting the response of the food industry to the obesity epidemic and other nutritional challenges for noncommunicable disease prevention. For that reason this and other work documenting changes in the nutritional quality of foods, now a leading determinant of the burden of chronic disease, is important.

Response: Thank you for this feedback. Below we provide specific responses to your comments.

2. One weakness of the article was that it failed to clearly contextualize nutrition policy initiatives that were occurring at the same time - these included not only the dissemination of menu labeling which began in 2008 and became nationally mandatory in 2018, but also the National Salt Reduction Initiative and the FDAs voluntary targets for sodium reduction, and the prohibition of trans fats, also beginning in NYC in 2007 and spreading nationally by 2015. This timing should be more clearly laid out.

Response: We have added the following text to the discussion:

“It is possible that the declines in calories and nutrients in this study are related to local or national nutrition policies; although this is unknowable from our data as we lack both pre and post implementation data for relevant policies. For example, the menu labeling rule included in the 2010 Affordable Care Act [14], was only implemented nationally on May 2018 (after several delays); the law mandates that calorie information be posted on menus and menu boards in restaurants or similar retail food establishments with more than 20 outlets. The data available for this study ends prior to the menu labeling implementation date. Prior to the Affordable Care Act, menu labeling regulations were adopted by more than 20 localities, beginning with New York City in 2008.[15] Another relevant policy is the National Salt Reduction Initiative (NSRI), which launched in 2009, a partnership with state and local health authorities and national health organizations that encouraged food manufacturers and restaurants to voluntarily reduce sodium in their products.[16, 17] A third policy is the ban on trans fats in U.S. restaurants and grocery stores which was announced by the Food and Drug Administration in 2015 (giving food-makers three years to phase out the ingredient).[18]”

3. I was also unclear why the year of adoption of menu labeling was added to the modeling, and did not see the clear presentation of analysis of that variable in the adjusted model.

Response: Prior research by our group has shown that restaurants that post nutrition information have fewer per-item calories on their menus (Bleich et al Health Affairs 2015). Therefore, we felt it was important to adjust for the menu labeling status of the restaurant included in our analysis. To this point, we already include the below text. If the editor feels that this is unclear, we are happy to revise. 

“Given that many large chain restaurants voluntarily implemented menu labeling in advance of the federal rule and during the study period,[8] we additionally included an indicator for the year the restaurant began labeling their menus with calories.”

4. In general it would be helpful if the tables included not just the change in grams of each nutrient or calories but in %. This facilitates understanding whether the changes were significant not just statistically but practically. The decline in calories was 25% in new items, which is actually quite substantial.

Response: We agree that the percent change is helpful and already include that for all significant findings that are reported in the text. We worry that adding the percent change in the tables might make them very busy. So, our preference is to leave as is, but if the editor feels strongly about this we are happy to make this change.

5. In regards to interpretation of any increase in saturated fat, it would be helpful to also display trans fat and trans fat + saturated fat. The national ban on partially hydrogenated oils went into effect during the period of the study. While many chains had already gotten rid of artificial trans fat a quick perusal of MenuStat suggests tat some were still using PHOs. Ruminant trans fat of course continues present. But if the goal was for trans fat to be replaced by healthier fats to the extent possible, ideally that total of Trans + saturated fats would decline even if saturated fat increased slightly, since saturated were needed to some extent to replace PHOs, particularly in baking applications.

Response: We agree that this an interesting and important question. We have updated all relevant tables to include this information. We find no significant changes in trans-fat for any menu category or subcategory when comparing our first year of data (2012 for common and 2013 for new) to data in 2018, and find no significant linear trend for this nutrient. 

6. In limitations, the absence of sufficient information on portion size should also be noted, which would have helped to interpret whether these changes represent less calorie dense foods or just less food. This requirement is not part of mandatory nutrition information under ACA. As noted much of the change in nutrients reflected overall calorie content.

Response: We have added the following text to the limitations section:

“Sixth, because we lack complete and consistent data on volume across all the MenuStat data, we are unable to examine changes in portion size over time. 

7. S1 Table - It would be helpful if the table also provided the year menu labeling was added by each company

Response: Good suggestion. We have added that in. 

References

1. Bleich SN, Wolfson JA, Jarlenski MP. Calorie changes in large chain restaurants: declines in new menu items but room for improvement. American journal of preventive medicine. 2016;50(1):e1-e8.

2. Jarlenski MP, Wolfson JA, Bleich SN. Macronutrient composition of menu offerings in fast food restaurants in the US. American journal of preventive medicine. 2016;51(4):e91-e7.

3. Wolfson JA, Moran AJ, Jarlenski MP, Bleich SN. Trends in Sodium Content of Menu Items in Large Chain Restaurants in the US. American journal of preventive medicine. 2018;54(1):28-36.

4. Bleich SN, Wolfson JA, Jarlenski MP. Calorie changes in chain restaurant menu items: implications for obesity and evaluations of menu labeling. American journal of preventive medicine. 2015;48(1):70-5.

5. Frelier JM, Moran AJ, Vercammen KA, Jarlenski MP, Bleich SN. Trends in Calories and Nutrients of Beverages in US Chain Restaurants, 2012–2017. American journal of preventive medicine. 2019;57(2):231-40.

6. Bleich SN, Wolfson JA, Jarlenski MP, Block JP. Restaurants with calories displayed on menus had lower calorie counts compared to restaurants without such labels. Health affairs. 2015;34(11):1877-84.

7. Moran AJ, Block JP, Goshev SG, Bleich SN, Roberto CA. Trends in nutrient content of children’s menu items in US chain restaurants. American journal of preventive medicine. 2017;52(3):284-91.

8. Cleveland LP, Simon D, Block JP. Compliance in 2017 With Federal Calorie Labeling in 90 Chain Restaurants and 10 Retail Food Outlets Prior to Required Implementation. American journal of public health. 2018;108(8):1099-102. Epub 2018/06/22. doi: 10.2105/ajph.2018.304513. PubMed PMID: 29927646; PubMed Central PMCID: PMCPMC6050842.

9. Hills AP, Byrne NM, Lindstrom R, Hill JO. 'Small changes' to diet and physical activity behaviors for weight management. Obesity facts. 2013;6(3):228-38. Epub 2013/05/29. doi: 10.1159/000345030. PubMed PMID: 23711772; PubMed Central PMCID: PMCPMC5644785.

10. Jensen MD, Ryan DH, Apovian CM, Ard JD, Comuzzie AG, Donato KA, et al. 2013 AHA/ACC/TOS guideline for the management of overweight and obesity in adults: a report of the American College of Cardiology/American Heart Association Task Force on Practice Guidelines and The Obesity Society. Circulation. 2014;129(25 Suppl 2):S102-38. Epub 2013/11/14. doi: 10.1161/01.cir.0000437739.71477.ee. PubMed PMID: 24222017; PubMed Central PMCID: PMCPMC5819889.

11. Raynor HA, Champagne CM. Position of the Academy of Nutrition and Dietetics: Interventions for the Treatment of Overweight and Obesity in Adults. J Acad Nutr Diet. 2016;116(1):129-47. Epub 2016/01/01. doi: 10.1016/j.jand.2015.10.031. PubMed PMID: 26718656.

12. Urban LE, McCrory MA, Dallal GE, Das SK, Saltzman E, Weber JL, et al. Accuracy of stated energy contents of restaurant foods. Jama. 2011;306(3):287-93. Epub 2011/07/21. doi: 10.1001/jama.2011.993. PubMed PMID: 21771989; PubMed Central PMCID: PMCPMC4363942.

13. Technomic. Top 500 Chain Restaurant Report 2019 [cited 2019 December 11]. Available from: https://www.technomic.com/available-studies/industry-reports/top-500.

14. Patient protection and affordable care act, Pub. L. No. 48(2010).

15. Center for Science in the Public Interest. State and Menu Labeling Policies Washington, DC2014 [cited 2019 December 10]. Available from: https://cspinet.org/resource/state-and-menu-labeling-policies.

16. New York City Department of Health and Mental Hygiene. National Salt Reduction Initiative Corparate Achievements 2014 [cited 2019 Devember 10]. Available from: https://www1.nyc.gov/assets/doh/downloads/pdf/cardio/nsri-corporate-commitments.pdf.

17. New York City Department of Health and Mental Hygiene. Sodium Initiatives 2018 [cited 2019 December 10]. Available from: https://www1.nyc.gov/site/doh/health/health-topics/national-salt-reduction-initiative.page.

18. U.S. Food and Drug Administration. Final Determination Regarding Partially Hydrogenated Oils (Removing Trans Fat) 2018 [cited 2019 December 10]. Available from: https://www.fda.gov/food/food-additives-petitions/final-determination-regarding-partially-hydrogenated-oils-removing-trans-fat.

---

## [Editor Report · Decision Letter 1]

27 Jan 2020

Calorie and nutrient trends in large U.S. chain restaurants, 2012-2018

PONE-D-19-29367R1

Dear Dr. Bleich,

We are pleased to inform you that your manuscript has been judged scientifically suitable for publication and will be formally accepted for publication once it complies with all outstanding technical requirements.

With kind regards,

David Meyre

Academic Editor

PLOS ONE
---

## [Editor Report · Acceptance letter]

30 Jan 2020

PONE-D-19-29367R1 

Calorie and nutrient trends in large U.S. chain restaurants, 2012-2018 

Dear Dr. Bleich:

I am pleased to inform you that your manuscript has been deemed suitable for publication in PLOS ONE. Congratulations! Your manuscript is now with our production department. 

With kind regards,

on behalf of

Dr David Meyre 

Academic Editor

PLOS ONE